

# Reconstructing hadronically decaying tau leptons with a jet foundation model

**Laurits Tani[1,2⋆], Joosep Pata[1†] and Joschka Birk[3‡]**

**1** National Institute Of Chemical Physics And Biophysics (NICPB),
Rävala pst. 10, 10143 Tallinn, Estonia
**2** Istituto Nazionale di Fisica Nucleare Sezione di Bari, Via E. Orabona n.4, I-70126 Bari, Italy
**3** Institute for Experimental Physics, Universität Hamburg,
Luruper Chaussee 149, 22761 Hamburg, Germany

⋆ laurits.tani@cern.ch , † joosep.pata@cern.ch , ‡ joschka.birk@uni-hamburg.de

## Abstract

The limited availability and accuracy of simulated data has motivated the use of foundation models in high energy physics, with the idea to first train a task-agnostic model on large and potentially unlabeled datasets. This enables the subsequent fine-tuning of the learned representation for specific downstream tasks, potentially requiring much smaller datasets to achieve performance comparable to models trained from scratch on larger datasets. We study how OmniJet-$\alpha$, one of the proposed foundation models for particle jets, can be used on a new set of tasks, and on a new dataset, in order to reconstruct hadronically decaying $\tau$ leptons. We show that the pretraining can successfully be utilized for this multi-task problem, improving the resolution of momentum reconstruction by about 50% when the pretrained weights are fine-tuned, compared to training the model from scratch. While much work remains ahead to develop generic foundation models for high-energy physics, this early result of generalizing an existing model to a new dataset and to previously unconsidered tasks highlights the importance of testing the approaches on a diverse set of datasets and tasks.

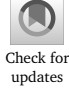

# 1 Introduction

Pretrained models that can be adapted for several tasks have recently emerged as a promising approach to reduce the amount of data required for supervised learning tasks in particle physics [1, 2]. Such adaptable models have also been called foundation models (FMs) [3], based on the corresponding research from natural language processing and computer vision. The underlying principle involves pretraining a model on a large dataset using a generic task. This pretraining allows the model to learn generalizable intermediate representations or embeddings that can then be used for downstream physics tasks. The model is subsequently fine-tuned on specific tasks, often demonstrating that pretraining significantly reduces the amount of required data to obtain the same performance as the corresponding non-pretrained model. In this paper, we investigate the transferability of an existing pretrained foundation model to a new dataset (out-of-domain) and a new set of tasks (out-of-context), extending the model from jet generation and jet tagging to machine learning-based hadronically decaying tau ($\tau_{\mathrm{h}}$) reconstruction.

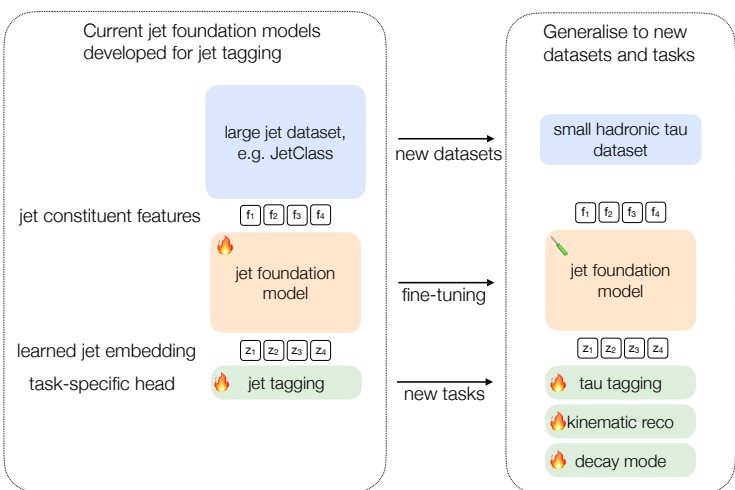

Figure 1: Left: the current typical workflow for using jet-based foundation models. The FM is first trained on a self-supervised task using a large pre-existing dataset such as JetClass, followed by fine-tuning on a specific tagging task on a dataset of similar type. The jet model takes as input the jet constituent feature vectors, and produces a jet embedding. Right: we demonstrate that jet foundation model generalizes to new tasks and new datasets. The weights of the foundation model can be frozen or fine-tuned, while the task-specific heads are optimized on a smaller full simulation dataset. We denote a full training from scratch with the fire symbol, and fine-tuning with the screwdriver symbol.

Table 1: Training types and their corresponding loss functions for the tasks that make up $\tau_{\mathrm{h}}$ identification and reconstruction.

| Task | Training type | Loss function |
|---|---|---|
| $\tau_{\mathrm{h}}$ identification | binary classification | focal loss |
| $p_{\mathrm{T}}$ reconstruction | regression | Huber loss |
| decay mode identification | multi-class classification | cross entropy loss |

Table 2: Features used for training the models for the three tau reconstruction and identification tasks.

| Category | Variable | Definition |
|---|---|---|
| Kinematics | $p_T^{cand}$ | transverse momentum of the jet constituents |
| | $m^{cand}$ | mass of the jet constituents |
| | $\Delta\eta$ | difference in $\eta$ between jet axis and jet constituents |
| | $\Delta\phi$ | difference in $\phi$ between jet axis and jet constituents |
| PID | charge | charge of the jet constituents |
| | isElectron | if the jet constituent is an electron |
| | isMuon | if the jet constituent is a muon |
| | isPhoton | if the jet constituent is a photon |
| | isChargedHadron | if the jet constituent is a charged hadron |
| | isNeutralHadron | if the jet constituent is a neutral hadron |

Currently available jet-based foundation models include OmniLearn [4], RS3L [1], OmniJet-$\alpha$ [5] and Masked Particle Modeling (MPM) [6,7]. The models can be subdivided based on the pretraining task into label-supervised (OmniLearn, RS3L) and self-supervised (OmniJet-$\alpha$, MPM) models that do not require labels from simulation. Label-supervised models rely on a well-established physics task such as jet classification in the pretraining phase. These models can be trained on large simulated datasets where labels (i.e. the flavor of the originator particles of each jet) are available based on simulation information. On the other hand, self-supervised models can be trained on unlabeled data, thus also on real experimental data [8], where the flavor of the originator particle of the jet is not known.

The aforementioned self-supervised models for jet physics employ either autoencoding (MPM) or autoregressive (OmniJet-$\alpha$) techniques to define the self-supervised pretraining objective. Autoencoding models reconstruct the original jet constituents after introducing a form of corruption, such as randomly removing or masking particles, while autoregressive approaches generally require discretizing the continuous particle features into tokens using a tokenizer that is trained beforehand, with vector quantized variational autoencoder (VQ-VAE) [5,6] being a popular choice in HEP. With the aim to test all the available jet foundation models for $\tau_{\mathrm{h}}$ identification and reconstruction, we opted to start with OmniJet-$\alpha$ in this study. While in this work we study the use of OmniJet-$\alpha$ for the task of hadronic tau reconstruction, the analysis of other approaches such as MPM is left for future work.

## 2 Hadronically decaying tau lepton reconstruction with a pretrained model

In this study, the OmniJet-$\alpha$ foundation model, pretrained on the JetClass dataset [9, 10] (a benchmark dataset for jet tagging algorithms), is applied to the task of hadronically decaying tau lepton ($\tau_h$) reconstruction and identification using the Fu$\tau$ure dataset [11]. Training and evaluating a model on the same dataset (JetClass) represents an in-domain scenario. In this work, we show how the OmniJet-$\alpha$ model generalizes by taking a model pretrained on JetClass and applying it to the Fu$\tau$ure dataset, which differs in physics process (proton-proton to electron-positron), simulation granularity (Delphes [12] $\rightarrow$ full simulation and reconstruction), and center-of-mass energy, thus representing an out-of-domain scenario. Moreover, the training tasks differ significantly between the two trainings – in addition to jet tagging, reconstructing $\tau_h$ also involves out-of-context tasks such as kinematic and decay mode reconstruction. Although training both the backbone and tokenizer on the Fu$\tau$ure dataset could mitigate the out-of-domain effects, it would not address the challenge of generalizing to new tasks.

The machine learning targets in the $\tau_h$ reconstruction task are defined by matching the reconstructed jets to generator-level hadronic tau leptons following the approach in [11], described briefly below. First, both generator-level stable particles and reconstructed particles are clustered using the generalized-$k_t$ algorithm for electron-positron events [13] with $p = -1$ and $R = 0.4$ using FastJet [14]. Jets within $\Delta R < 0.4$ of a generator-level electron or muon are removed from the dataset to create a cleaner reference sample that is more representative of the hadronic $\tau$ decay. Second, we select generator-level hadronically decaying tau leptons based on the descendants of the tau lepton from the Pythia decay tree. Next, these generator-level hadronically decaying tau particles are matched to generator-level jets with a $p_T^{gen-jet} \geq 20$ GeV, with each jet having at most one matched hadronic tau particle. Subsequently, reconstructed jets are also matched to generator-level jets using the $\Delta R$ criterion, allowing us to make triplets of the generator-level tau lepton, the generator-level jet and the reconstructed jet.

In background samples, and in cases where the generator-level hadronic tau cannot be matched to a reconstructed jet, we assign the reconstructed jet to the background class. The training target for the $\tau_h$ identification task is a binary label indicating whether the jet was matched to a generator-level $\tau_h$ decay. If the jet was matched to a generator-level $\tau_h$, we also assign the true decay mode and visible $p_T$ from the generator-level tau lepton, as described below.

We consider a total of six $\tau_h$ classes for the decay mode reconstruction based on the number of charged ($h^\pm$) and neutral ($h^0$) hadrons: $h^\pm \nu_\tau$, $h^\pm h^0 \nu_\tau$, $h^\pm \geq 2h^0 \nu_\tau$, $h^\pm h^\mp h^\pm \nu_\tau$, $h^\pm h^\mp h^\pm \geq 1h^0 \nu_\tau$ and "rare", the latter containing all other $\tau_h$ decays not falling into any of the other five categories. These categories are assigned by inspecting the generator-level decay tree and the descendants of the hadronically decaying tau lepton, counting the charged and neutral descendants of the $\tau$ decay. These categories are the most prominent decay channels, and their precise experimental determination is important particularly for $\tau$ branching fractions and spin polarization measurements [15–17].

For the kinematic reconstruction of $\tau_h$ candidates, we use the visible $\tau_h$ $p_T$ as the training target, framing the kinematic reconstruction of the true $\tau_h$ momentum as a regression task. Accurately reconstructing the momentum of the $\tau_h$ from the decay products is an important input to analyses featuring $\tau$ leptons and $\tau$ polarization measurements [15], for example.

The OmniJet-$\alpha$ backbone, as indicated by "jet foundation model" in Fig. 1, consists of the encoding table, and three generative pre-trained transformer (GPT) [18] layers. A separate head is defined for each training task. Inspired by the ParticleTransformer architecture [9], each head consists of two multi-head attention (MHA) [19] layers and two class-attention

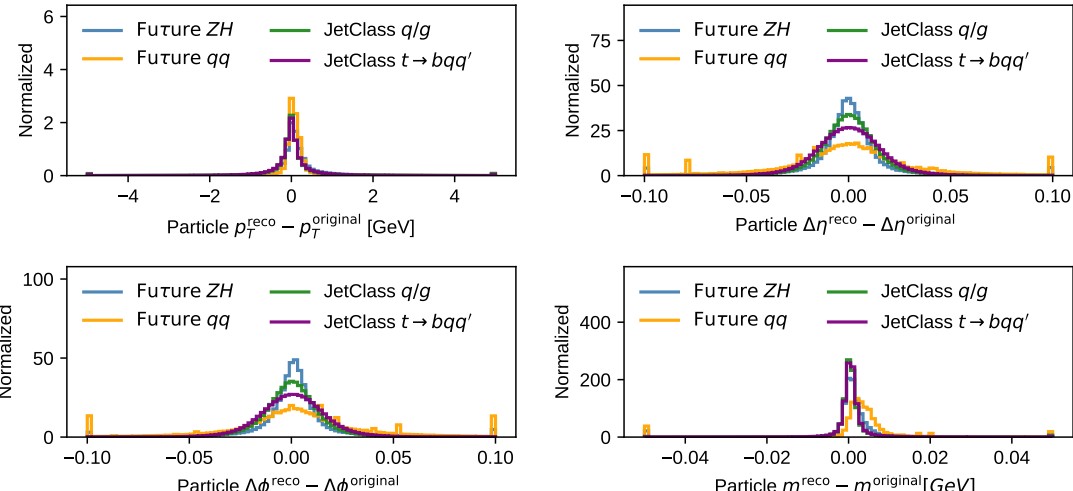

Figure 2: Differences of the reconstructed and original values for JetClass and Fuτure datasets after tokenization and the decoder model, as a cross-check of the tokenization process. Despite being trained on a different dataset, we find that the tokenizer does not exhibit mode collapse in the Fuτure dataset and is generally able to represent the jets from $\tau_h$ decays. The spikes in the $Z/\gamma^* \to qq$ distribution are caused by jets with only one constituent, which results in a discrete component of the reconstructed distributions.

blocks [9, 20]. For both the VQ-VAE and backbone training we switched to the Ranger optimizer [21–23] as it resulted in more stable training behavior. The backbone model was trained for 7 epochs on next-token prediction using 20 million quark/gluon and $t \to bqq'$ jets from the JetClass dataset. Apart from these minor modifications, the architecture and overall training strategy for the VQ-VAE and backbone remain consistent with those described in [5].

Following the variable selection choices made in the original OmniJet-$\alpha$ [5] for selecting input variables, a slightly modified set of features from Ref. [9] was used. This modified set of features excludes the $p_T$ and $E$ of the candidates relative to the corresponding jet variables, as well as the $\Delta R$ of the candidate relative to the jet axis, but includes their mass.

To incorporate additional variables not included in Ref. [5], we retrained both the tokenizer and the backbone model on JetClass using the variables in Tab. 2. Extending the variable set by the additional particle identification (PID) features required increasing the tokenizer codebook size from 8 192 to 32 000 tokens to maintain a good resolution of the kinematic features reconstructed from the tokens. This increase in codebook size directly impacts the number of parameters in the backbone model, as it scales with the number of tokens. However, we did not observe significant changes in the backbone training dynamics as a result of this increase. We achieved a stable VQ-VAE [24, 25] training by slightly increasing the latent space dimension to 8 (from 4) and setting a patience of 100 (instead of 10) before replacing unused tokens.

The trajectory displacement variables listed in Ref. [9], also known as lifetime variables, can be important for $\tau$ reconstruction in certain scenarios. Similarly to the case of b-hadrons, the finite lifetime of $\tau$ leptons causes the charged decay products to exhibit a measurable displacement between the primary and secondary vertices. Given the $\tau$ lifetime of $\sim$290 fs, a typical $\tau$ lepton with 30 GeV of energy travels about $\sim$2 mm before decaying. This displacement provides useful information for distinguishing jets from $\tau$ decays and those originating from quarks or gluons. While utilization of lifetime variables would likely lead to improve-

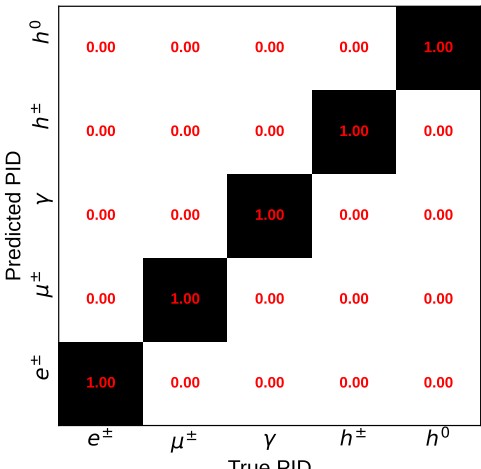 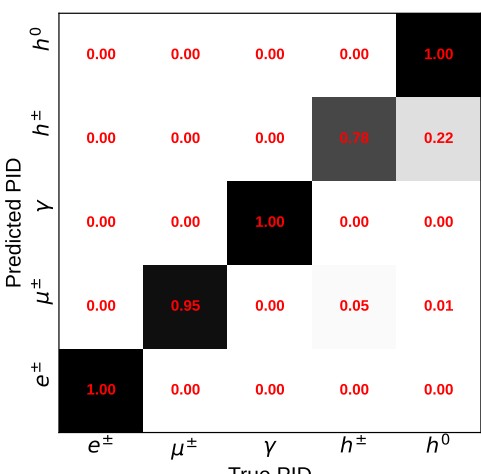

Figure 3: Reconstructing the PID from the JetClass (left) and Fuτure (right) dataset using the encoder and decoder from the VQ-VAE tokenizer, normalized by reconstructed values. We find that in terms of the categorical variables the tokenizer and decoder reconstruct the original JetClass training dataset accurately, but confuse charged and neutral hadrons. As we do not observe further pathological behavior such as complete mode collapse when moving to a new dataset, we find that the tokenizer is suitable for the subsequent studies.

ment [6], especially in tau tagging performance, these features are challenging to integrate into the VQ-VAE-based tokenization scheme due to the large tails of the corresponding distributions, and are therefore excluded from this study.

Efficiently representing complex feature distributions in the tokenizer, and the necessity of the tokenization itself, are topics that are actively being studied for the development of jet-based foundation models [7].

## 3 Results

### 3.1 Dataset tokenization and pretraining

The token reconstruction performance on both JetClass and Fuτure datasets is shown in Fig. 2. In addition to the differences in simulation accuracy and transverse momenta of the jets in JetClass and Fuτure datasets, the poorer performance in reconstructing $Z \to qq$ jet tokens may be due to their increased collimation at these energies. Additionally, the jet mass distributions differ between the two samples: $Z \to qq$ jets have a mass of 91 GeV, while top jets have a mass of 173 GeV. These differences likely affect the reconstruction quality and resolution of the jet constituents, as the tokenizer struggles to generalize to lower-energy samples.

Nevertheless, the resolution for the signal sample remains higher than that of the JetClass samples used during tokenizer training. While categorical features such as the PID are accurately reconstructed for the JetClass dataset as shown in Fig. 3, this is not the case for the Fuτure dataset, where the neutral and charged hadrons are often confused. However, we do not observe pathological behavior such as mode collapse in any of the features, highlighting that the tokenizer is able to represent jets from a significantly different dataset, and is thus generalizable to new domains.

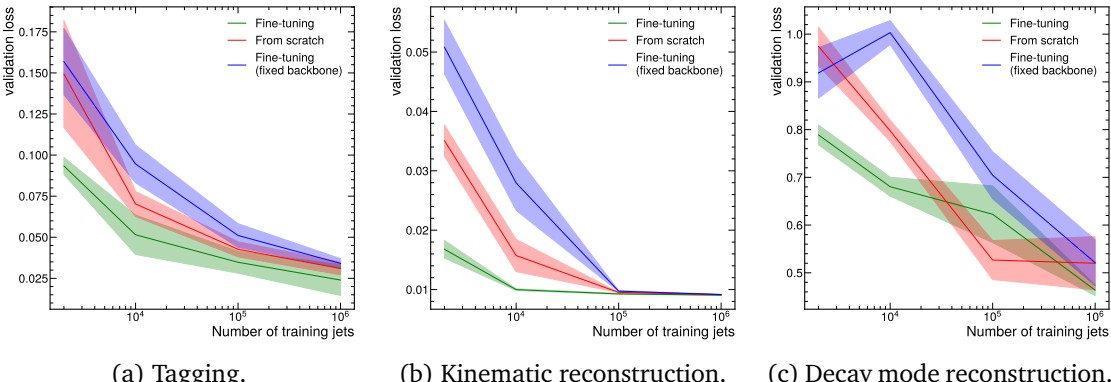

| (a) Tagging. | (b) Kinematic reconstruction. | (c) Decay mode reconstruction. |

Figure 4: We compare the performance of the three training strategies on the three $\tau_h$ reconstruction tasks as a function of the training dataset size based on the smallest validation loss. We see that the case of fine-tuning both the pretrained weights and the head yields the best performance, significantly outperforming the training from scratch or with a fixed backbone at small training dataset sizes.

## 3.2 Fine-tuning to $\tau_h$ reconstruction tasks

We train the $\tau_h$ reconstruction and identification models for a total of 100 epochs using 4096 jets per batch. The training types and their corresponding loss functions for each task are given in Tab. 1, with the exact settings of each loss function being detailed in Refs. [26] and [11].

Following established practice in HEP foundation model research, we study the performance in low training data regimes by varying the training dataset size from $\simeq 10^3$ to $\simeq 10^6$ jets and evaluating the performance of the model on three different machine learning tasks that make up hadronically decaying tau reconstruction and identification.

The validation dataset comprised approximately 540,000 $\tau_h$ jets for $\tau_h$ decay mode and kinematic reconstruction, and 1.68 million $\tau_h$ (signal) and q/g (background) jets for the tau tagging task. The testing dataset included around 860,000 jets for the former and 2.29 million for the latter. To study the effectiveness of the pretrained backbone model, we evaluated three distinct training strategies.

In the "from scratch" strategy, we train the backbone model from a random weight initialization. This scenario provides a baseline performance for the model architecture and training strategy on the downstream task. To mimic a typical transfer learning [27–29] application, we initialize the backbone model with the pretrained weights from the generative pretraining on JetClass. In this strategy, we allow the weights of the backbone to be optimized midway during training. The choice of epoch for unfreezing the pretrained weights is a tunable hyperparameter, which we discuss further in Sec. 3.4. This training strategy is referred to as "fine-tuning". To assess the possible benefit of the pretrained backbone, we evaluate the representations learned during pretraining by freezing all weights in the backbone model and keeping them fixed. This training strategy is referred to as "fixed backbone".

We leave the tokenizer weights frozen for all three strategies. While allowing them to be updated during training might help mitigate out-of-domain effects, this was not a consideration in Ref. [5], where only in-domain data was used. Since our goal is not to develop the foundation model itself, but rather to demonstrate that existing foundation models can be adapted to different datasets and tasks, we follow the same strategy and keep the parameters frozen during downstream training to minimize the differences with respect to [5].

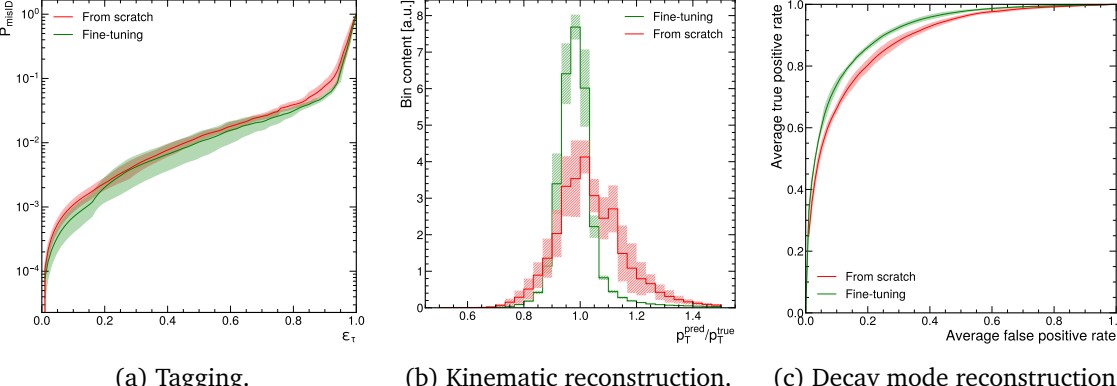

(a) Tagging.      (b) Kinematic reconstruction.      (c) Decay mode reconstruction.

Figure 5: We compare fine-tuning the OmniJet-$\alpha$ model (green) with training it from scratch (red) based on physics metrics for the three $\tau_\mathrm{h}$ identification and reconstuction tasks. The training used $10^4$ jets, corresponding to ∼1% of the full dataset, showing the mean and variation across three independent trainings. (a) For $\tau_\mathrm{h}$ at 80% efficiency fine-tuning improves the average mis-identification ($P_\mathrm{misID}$) rate by 20% (b) We see that fine-tuning the model from pre-existing weights improves the resolution of the $p_\mathrm{T}$ reconstruction by ∼50% on small training datasets compared to training the model from scratch. (c) Fine-tuning the OmniJet-$\alpha$ model for decay mode reconstruction shows an improvement of ∼3% in AUC of over training the model from scratch. Note that for this multiclass classification task, we employed a one-vs-rest approach and calculated the AUC score as an average across all decay modes, with each class weighted according to the proportion of jets in that class.

## 3.3 General performance

We evaluate the model based on a total of three trainings that are performed for each $\tau$ reconstruction task, each training strategy and three dataset sizes. In Fig. 4 we compare the validation losses of the three tasks. We see that fine-tuning the backbone almost always achieves the best performance across all training tasks, with the performance differences between the three strategies diminishing as the training dataset size increases. On the other hand, the fixed backbone strategy performed the worst in comparison to the other two strategies. This could arise from several factors, including the physics differences between the jets in the JetClass ($pp \rightarrow q/g/t/H/W/Z$ and Fu$\tau$ure datasets (ee $\rightarrow q/g/\tau$), and the granularity of the simulation and reconstruction between the two datasets (Delphes vs. full Geant4 simulation with full reconstruction), resulting in the backbone weights trained on JetClass to require significant adaptation to be useful on the Fu$\tau$ure dataset.

In addition to comparing validation losses, evaluations were also conducted using more physically motivated metrics - AUC score for $\tau$ tagging and decay mode reconstruction, and interquartile range (IQR) to measure momentum resolution for kinematic reconstruction. Fig. 5 illustrates the possible physics improvements of using a pretrained backbone over training the model from scratch for a training using $10^4$ jets corresponding to ∼ 1% of the full dataset. The clear hierarchy in validation losses can also be observed when comparing physics performance of the training strategies. Firstly, comparing "fine-tuning" to "from scratch", we see a 20% improvement in mis-identification rate at 80% efficiency. Similarly, the $p_\mathrm{T}$ response distribution is approximately centered around 1 and is significantly narrower when fine-tuning the pretrained backbone, with an improvement in resolution of about 50% compared to a model trained from scratch, indicating that the pretraining provides useful information for this task, despite the differences in the datasets and the novelty of the task. We also observe a clear

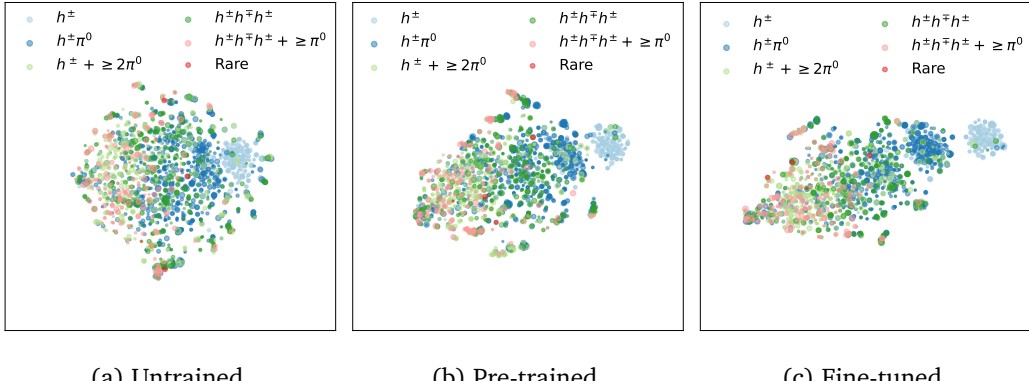

|  (a) Untrained. | (b) Pre-trained. | (c) Fine-tuned. |

Figure 6: The embeddings of 4096 jets from the backbone model, projected into two dimensions using the tSNE algorithm. Each marker represents one true $\tau_h$ jet, with the color of the marker representing the true decay mode of the $\tau_h$. We see that in terms of the true decay mode of the jet, the jet embeddings of the pretrained backbone (b) cluster in a more significant way than the untrained model (a). The backbone pretrained on the JetClass dataset results in a separation between the $h^{\pm}$ decay mode and the other decay modes, illustrating that the pretraining contains information relevant for a completely different task on a different dataset. In (c), we see that fine-tuning the backbone also results in the $h^{\pm}\pi^0$ decay mode being separated already at the level of the embedding, illustrating that the fine-tuning can further improve the relevance of the embeddings for this dataset and task.

benefit of fine tuning over training the model from scratch for decay mode reconstruction, where we see an improvement of about 3% in the AUC.

While using the pre-trained backbone provides useful information for the downstream tasks, we note that the supervised ParticleTransformer baseline trained specifically for each task generally outperforms both approaches that use OmniJet-$\alpha$. With $10^4$ training jets the ParticleTransformer model significantly outperforms the fine-tuning for tagging (70% lower mis-identification rate at the studied working point) and for decay mode classification (5% higher AUC), while the regression is somewhat better for the fine-tuning (30% narrower), with the ParticleTransformer gaining more rapidly in performance with the increased number of training jets than the fine-tuning. This highlights that while the pretraining is useful, the backbone does not always provide sufficient representational power for all tasks, when compared to the best-available supervised baseline. We believe that tokenization in the backbone may affect its performance. This is in line with recent findings on testing foundation model approaches in other scientific domains [30]. Studying and potentially improving the generalization capability of HEP foundation models remains an active direction of research.

Another way to evaluate the usefulness of the pretraining is to study the structure of the embedding space. For this we employ T-distributed Stochastic Neighbor Embedding (tSNE) algorithm [31] that maps high-dimensional data to a low-dimensional embedding, allowing us to visualize similarities of high-dimensional data samples. We compare the $\tau_h$ decay mode structure for three levels of training: untrained, pretrained on JetClass dataset, and fine-tuned. Fig. 6 demonstrates that the backbone model has learned something generic during pretraining, and shows that fine-tuning the model offers further improvement in the structure of the embedding space.

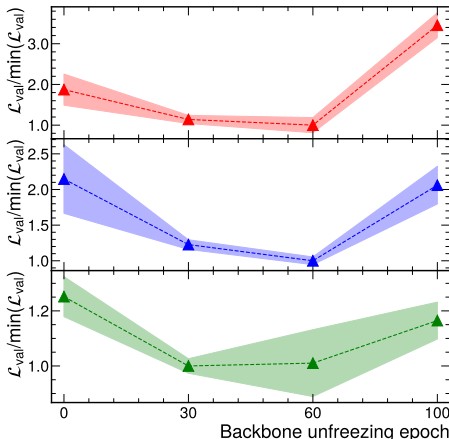 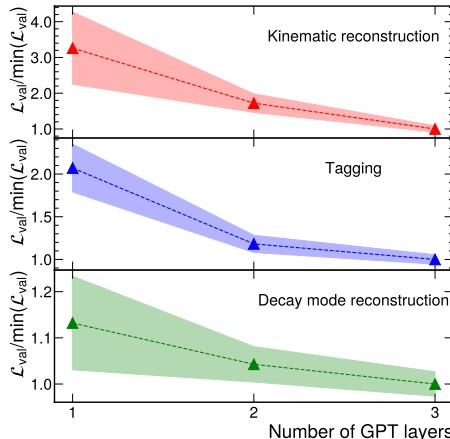

Figure 7: (Left) We study the effect of when the backbone is unfrozen during the fine-tuning. We find that the best results are achieved in the fine-tuning phase if the backbone is kept frozen for the first 60 of the 100 epochs. (Right) We also study how reducing the backbone model size by using a limited number of layers affects the downstream performance. We find that it is significantly better to use all pretrained GPT layers, as opposed to using only a few layers. We plot the validation loss on the y-axis, normalized by the minimum validation loss across all points. This ensures that the lowest validation loss on both figures is set to 1.

### 3.4 Strategies for reusing the pretrained weights

Determining useful ways of leveraging the pretrained backbone poses another challenge. For example, we have introduced a degree of freedom on when to unfreeze the backbone parameters. Moreover, it is possible to use only a subset of the pretrained layers to reduce the possibilities for overtraining in downstream tasks. Since the effects of these choices are expected to be more pronounced with smaller datasets, we performed ablation studies using a training sample of 2,000 jets using all three tasks.

The effects of unfreezing the backbone model parameters at four different points is shown in Fig. 7 (left). With a total of 100 epochs allocated for downstream task training, unfreezing at epoch 100 is equivalent to the fixed backbone strategy. Our results indicate that keeping the backbone weights unchanged for about half of the epochs leads to improved performance in comparison to unfreezing from start or keeping the backbone weights fixed.

Furthermore, our ablation studies suggest that using all pretrained GPT blocks results in the best performance (Fig. 7 (right)), indicating that successive layers learn more useful representations in pretraining that can be leveraged for downstream tasks.

## 4 Summary and outlook

In this paper, we applied OmniJet-$\alpha$, a foundation model pretrained on the JetClass dataset, in an out-of-context and out-of-domain manner for machine learning based hadronically decaying $\tau$ lepton reconstruction and identification on a full simulation and reconstruction dataset.

We demonstrated that the generative pretraining on JetClass improves the performance on downstream hadronically decaying $\tau$ lepton reconstruction tasks on a different dataset, with more significant improvements for smaller training datasets. This means that in addition to being successfully used for new tasks that were not considered during pretraining, these foundation models can be extended for use in different physics processes.

Furthermore, across all three tasks, we find that due to the significant change in the dataset and the tasks, fine-tuning the foundation model considerably improves the performance compared to keeping the pretrained weights fixed. These conclusions are the most noticeable in the context of kinematic reconstruction, specifically in the hadronically decaying $\tau$ lepton $p_{\mathrm{T}}$ response distribution, where we observe that the improvement in resolution from fine-tuning is approximately $\sim 55\%$.

By studying the structure of the embedding space, we find that the learned jet embeddings are more correlated with decay mode structure after pretraining on JetClass and even more correlated with the decay mode after the task-specific fine-tuning, demonstrating both that the backbone model has learned something generic from the pretraining, and that further fine-tuning can improve the structure of the embedding space.

We also studied different strategies for reusing the pretrained weights, and find that the way the pretrained model is used significantly affects the performance of the downstream tasks. In particular, we find that in the fine-tuning scenario, the final performance on all tasks is considerably improved when the pretrained weights are fixed for about half of the training steps, while the task-specific heads should be left floating throughout.

Much work remains ahead to fully map the potential savings achievable with high energy physics foundation models. One of the key benefits of foundation models is the ability to use unlabeled, real data for pretraining, which could allow to address the possible simulation to reality gap. Therefore, the next logical step would be to perform the FM pretraining on existing real datasets, such as historical or current open data from colliders. For this, efforts have already been made using CMS Open Data [8], which we aim to extend in the future studies by incorporating the historical electron-positron collision data from the ALEPH experiment. Ongoing work to convert the ALEPH collision data into a more recent HEP data format is ongoing [32].

As OmniJet-$\alpha$ is only one foundation model among several others proposed for high energy physics in a rapidly developing field, the ability of such models to generalize to a wide variety of tasks including reconstructing and identifying hadronically decaying $\tau$ leptons will need to be studied in future work.

## Acknowledgments

We would like to thank Gregor Kasieczka, Anna Hallin, Michael Kagan and Hardi Vanaveski for discussions during the preparation of this study and comments on the manuscript.

**Funding information**    This work has been supported by the Estonian Research Council grants PSG864, RVTT3-KBFI and by the European Regional Development Fund through the CoE program grant TK202. LT was additionally supported by the MAECI grant. JB acknowledges support by the Deutsche Forschungsgemeinschaft under Germany's Excellence Strategy – EXC 2121 Quantum Universe – 390833306. This research was supported in part through the Maxwell computational resources operated at Deutsches Elektronen-Synchrotron DESY, Hamburg, Germany.

**Data availability**    The dataset used in this paper is made available in Ref. [33]. The software used to produce the results can be found in Ref. [34, 35].

**Author contributions**   **LT**: Conceptualization, Methodology, Software, Validation, Formal analysis, Investigation, Data Curation, Writing - Original Draft, Writing - Review & Editing, Visualization. **JP**: Conceptualization, Methodology, Software, Validation, Formal analysis, Investigation, Data Curation, Writing - Original Draft, Writing - Review & Editing, Visualization, Supervision, Project administration, Funding administration. **JB**: Conceptualization, Methodology, Software, Writing - Original Draft, Writing - Review & Editing.

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
