# Peer review of "Reconstructing hadronically decaying tau leptons with a jet foundation model"

_SciPost Physics Core, doi:SciPost Phys. Core 8, 046 (2025)_

## Round 1 · Referee Report · Anonymous (Referee 1) · 2025-5-6

Strengths

1- This paper demonstrates for the first time how foundation models can be transferred to a different dataset and an all new set of tasks on a new set of physics processes

2- The paper description, procedure and benchmarking is clearly defined and easy to follow

3- The background description and review is quite complete

Weaknesses

1- The only benchmarks for comparison of performance on the tau tasks are training the model from scratch, this means that a very large data hungry model is often trained on small datasets. It would be informative to know how a smaller model performs on smaller datasets

2- The paper makes several referrals to their “approach” which is a bit misleading I think as really the paper demonstrates that FMs can be applied to different datasets.

Report

This paper demonstrates for the first time how foundation models can be transferred to a different dataset and an all new set of tasks on a new set of physics processes. The paper is clearly written and has a clear set of downstream tasks, it also describes the relevant background clearly. At least one simple baseline model trained from scratch would be useful to improve the benchmarking and the wording in certain places could be made clearer. Overall the paper does a good job of exploring the use cases of foundation models to new datasets and tasks, demonstrating for the first time that they may actually be useful in a wide range of applications. I think the paper should be accepted after some small additions.

Requested changes

1- Abstract: “potentially requiring much smaller dataset sizes to reach the performance of models trained from scratch.” I do not think this is a complete sentence on its own, the implication is that smaller dataset sizes are needed relative to models trained from scratch on larger datasets, but this isn’t clear from the text. This is a confusing construction and should be made clearer.

2- Abstract: “and in a new dataset” —> “and on a new dataset”

3- Figure 1 caption: “the current typical workflow for training jet-based foundation models” to me this is really the current workflow for evaluating FMs

4- Figure 1 caption: “Right: we generalize the jet foundation model to new tasks and new datasets.” —> ““Right: we demonstrate that jet foundation model generalize to new tasks and new datasets.” The paper does not actually change the training or fine tuning process of FMs mechanically, all of the training loops remain the same. This is actually great because if the general overall training approach had to be updated for new datasets and tasks then the appeal of FMs would be reduced, as significant additional work would be required for each new dataset and task.

5- Introduction: “label-supervised (OmniLearn)” —> “label-supervised (OmniLearn, RS3L)” I think the resimulation paper current reference [1] should be included.

6- Section 2: “our approach involves taking a model pretrained on JetClass” I think this should be reworded to say that the authors show the model generalizes. I do not think there is a specific approach that can be referenced here.

7- Section 3.3: “backbone consistently achieves the best performance” Except for decay mode reconstruction at one point? This should probably be commented on or just changed to “almost always achieves the best performance”

8- General note on evaluation: It would be good to have a baseline for performance using something other than just from scratch. In particular it would be good to see how a smaller model behaves when trained from scratch. I personally question whether smaller models can be trained from scratch and improve the performance on smaller datasets. I’d be happy with any kind of reasonable downscaling of the base OmniJet-\alpha model applied to all dataset sizes, or some discussion in the text of why the large base model would be applied to such small datasets

9- Figure 6: Not sure about titles without capitals vs putting something in the caption. In general the figure fonts could be larger to more closely match the text.

Recommendation

Publish (easily meets expectations and criteria for this Journal; among top 50%)

  • validity: good
  • significance: good
  • originality: good
  • clarity: high
  • formatting: reasonable
  • grammar: reasonable

Author:  Laurits Tani  on 2025-05-23  [id 5514]

(in reply to Report 1 on 2025-05-06)

1- Abstract: “potentially requiring much smaller dataset sizes to reach the performance of models trained from scratch.” I do not think this is a complete sentence on its own, the implication is that smaller dataset sizes are needed relative to models trained from scratch on larger datasets, but this isn’t clear from the text. This is a confusing construction and should be made clearer.

    Reworded the sentence to make the point clearer

2- Abstract: “and in a new dataset” —> “and on a new dataset”

    Done

3- Figure 1 caption: “the current typical workflow for training jet-based foundation models” to me this is really the current workflow for evaluating FMs

    Replaced "training" with "using"

4- Figure 1 caption: “Right: we generalize the jet foundation model to new tasks and new datasets.” —> ““Right: we demonstrate that jet foundation model generalize to new tasks and new datasets.” The paper does not actually change the training or fine tuning process of FMs mechanically, all of the training loops remain the same. This is actually great because if the general overall training approach had to be updated for new datasets and tasks then the appeal of FMs would be reduced, as significant additional work would be required for each new dataset and task.

    Done.

5- Introduction: “label-supervised (OmniLearn)” —> “label-supervised (OmniLearn, RS3L)” I think the resimulation paper current reference [1] should be included.

    Done.

6- Section 2: “our approach involves taking a model pretrained on JetClass” I think this should be reworded to say that the authors show the model generalizes. I do not think there is a specific approach that can be referenced here.

    Agreed, changed the text accordingly

7- Section 3.3: “backbone consistently achieves the best performance” Except for decay mode reconstruction at one point? This should probably be commented on or just changed to “almost always achieves the best performance”

    Changed the text accordingly.

8- General note on evaluation: It would be good to have a baseline for performance using something other than just from scratch. In particular it would be good to see how a smaller model behaves when trained from scratch. I personally question whether smaller models can be trained from scratch and improve the performance on smaller datasets. I’d be happy with any kind of reasonable downscaling of the base OmniJet-\alpha model applied to all dataset sizes, or some discussion in the text of why the large base model would be applied to such small datasets

    Added a comparison with the ParT to the text. Using a smaller model (reducing the number of GPT layers) will reduce the performance, as shown on Fig. 7 (scaling with the number of GPT layers).

9- Figure 6: Not sure about titles without capitals vs putting something in the caption. In general the figure fonts could be larger to more closely match the text.

    Increased the font size in the legend and moved the text from title to caption.

From “weaknesses” section: 1) The paper makes several referrals to their “approach” which is a bit misleading I think as really the paper demonstrates that FMs can be applied to different datasets.

    All references to our "approach" have been reworded

2) The only benchmarks for comparison of performance on the tau tasks are training the model from scratch, this means that a very large data hungry model is often trained on small datasets. It would be informative to know how a smaller model performs on smaller datasets

    Added a comparison with the ParT to the text. Using a smaller model (reducing the number of GPT layers) will reduce the performance, as shown on Fig. 7 (scaling with the number of GPT layers).

---

## Round 1 · Referee Report · Anonymous (Referee 2) · 2025-5-9

Strengths

  1. The paper describes clearly the challenges of applying a Foundation Model (FM) on an out-of-domain dataset on a set of out-of-context tasks. The dataset features preprocessing, which is discussed in detail, and its shortcomings are discussed critically.
  2. The task described in the paper is realistic: the FM is applied on a very different dataset and distinct simulation settings, increasing the relevance of the findings.
  3. Authors show clear validations and have investigated in detail the training dynamics of the FM in various setups.
  4. The paper describes in detail the models, setups and training procedures.

Weaknesses

  1. A crucial part of the FM is the tokenisation model used to build the latent space representation of jet constituents. In the paper, the Authors tuned the original tokeniser from OmniJet-$\alpha$ to add some features important for the final task. Doing so, they mention they need to enlarge the size of the codebook considerably in order to obtain a good enough reconstruction error. Moreover, they mention that more features are difficult to incorporate into the tokeniser given their complex distribution. In my opinion, this is a pain point of the FM application, as the backbone model should include enough features to avoid the need to be retrained for different downstream tasks. Anyway, the authors stress this critical point in the paper: this is an experimental application of one of the first Jet FM models, and the selection of relevant features for a realistic, practical application will be crucial in future developments.

Report

The paper is one of the first applications of Foundation Models (FM) for High Energy Physics, showing an interesting application of Jet FMs to the challenging task of hadronic tau reconstruction. The out-of-domain task is explained clearly, and all the steps needed to adapt the original OmniJet-$\alpha$ model for this application are discussed in detail.

The results of the paper are very relevant for ML in HEP development and show promising performance for fine-tuning large backbone models for out-of-context tasks.

The validations and investigations performed by the authors on their results are complete and explained very clearly in the paper text.

I strongly recommend the publication of this paper.

Requested changes

  1. In the introduction in page 3, there is a paragraph about the MPMv1 backbone setup: I think it is quite dense and that it can be omitted as the paper is focusing on OmniJet-$\alpha$.

  2. Page 5: "from 8192 to 32000 token": This seems a large increase in the token codebook dimension. The authors should motivate this change more it and explain the consequences on the complexity and number of parameters in the backbone model. How does the codebook scale when more features need to be included in the object representation? Does the GPT backbone needs to become much larger to handle such larger codebook?

  3. Page 8 "We note that the supervised ParticleTransformer baseline trained specifically for each task outperforms both approaches of using OmniJet-α" how much it outperforms it? The authors should quantify the performance of a baseline model for an easier comparison, e.g. reporting the performance studied in Ref [11]

Recommendation

Publish (easily meets expectations and criteria for this Journal; among top 50%)

  • validity: top
  • significance: high
  • originality: top
  • clarity: top
  • formatting: excellent
  • grammar: excellent

Author:  Laurits Tani  on 2025-05-23  [id 5515]

(in reply to Report 2 on 2025-05-09)

1- In the introduction in page 3, there is a paragraph about the MPMv1 backbone setup: I think it is quite dense and that it can be omitted as the paper is focusing on OmniJet-α

    Removed the detailed description of MPM.

2- Page 5: "from 8192 to 32000 token": This seems a large increase in the token codebook dimension. The authors should motivate this change more it and explain the consequences on the complexity and number of parameters in the backbone model. How does the codebook scale when more features need to be included in the object representation? Does the GPT backbone needs to become much larger to handle such larger codebook?

    Added more details regarding the impact of the increased number of tokens.

3- Page 8 "We note that the supervised ParticleTransformer baseline trained specifically for each task outperforms both approaches of using OmniJet-α" how much it outperforms it? The authors should quantify the performance of a baseline model for an easier comparison, e.g. reporting the performance studied in Ref [11]

    Added a quantitative comparison between ParT and fine-tuned OmniJet-α for a specific dataset size for all tasks in the paragraph “While using the pre-trained backbone…”.

---

## Round 2 · Referee Report · Anonymous (Referee 2) · 2025-5-27

Report

The version 2 of the submission answers the questions raised in the previous Report and clarify my questions. The relevant sections of the paper have been improved, further enhancing the clarity of an already well-written manuscript.

I recommend the publication of the paper without any further revisions.

Recommendation

Publish (easily meets expectations and criteria for this Journal; among top 50%)

---

## Round 2 · Referee Report · Anonymous (Referee 1) · 2025-5-29

Report

The authors have addressed all of the points raised in the first reports and the manuscript is now acceptable.

Recommendation

Publish (meets expectations and criteria for this Journal)

---

## Round 2 · List of Changes

Addressed the following comments from reviewer #1:

1- Abstract: “potentially requiring much smaller dataset sizes to reach the performance of models trained from scratch.” I do not think this is a complete sentence on its own, the implication is that smaller dataset sizes are needed relative to models trained from scratch on larger datasets, but this isn’t clear from the text. This is a confusing construction and should be made clearer.

Reworded the sentence to make the point clearer

2- Abstract: “and in a new dataset” —> “and on a new dataset”

Done

3- Figure 1 caption: “the current typical workflow for training jet-based foundation models” to me this is really the current workflow for evaluating FMs

Replaced "training" with "using"

4- Figure 1 caption: “Right: we generalize the jet foundation model to new tasks and new datasets.” —> ““Right: we demonstrate that jet foundation model generalize to new tasks and new datasets.” The paper does not actually change the training or fine tuning process of FMs mechanically, all of the training loops remain the same. This is actually great because if the general overall training approach had to be updated for new datasets and tasks then the appeal of FMs would be reduced, as significant additional work would be required for each new dataset and task.

Done.

5- Introduction: “label-supervised (OmniLearn)” —> “label-supervised (OmniLearn, RS3L)” I think the resimulation paper current reference [1] should be included.

Done.

6- Section 2: “our approach involves taking a model pretrained on JetClass” I think this should be reworded to say that the authors show the model generalizes. I do not think there is a specific approach that can be referenced here.

Agreed, changed the text accordingly

7- Section 3.3: “backbone consistently achieves the best performance” Except for decay mode reconstruction at one point? This should probably be commented on or just changed to “almost always achieves the best performance”

Changed the text accordingly.

8- General note on evaluation: It would be good to have a baseline for performance using something other than just from scratch. In particular it would be good to see how a smaller model behaves when trained from scratch. I personally question whether smaller models can be trained from scratch and improve the performance on smaller datasets. I’d be happy with any kind of reasonable downscaling of the base OmniJet-\alpha model applied to all dataset sizes, or some discussion in the text of why the large base model would be applied to such small datasets

Added a comparison with the ParT to the text. Using a smaller model (reducing the number of GPT layers) will reduce the performance, as shown on Fig. 7 (scaling with the number of GPT layers).

9- Figure 6: Not sure about titles without capitals vs putting something in the caption. In general the figure fonts could be larger to more closely match the text.

Increased the font size in the legend and moved the text from title to caption.

From “weaknesses” section: 1) The paper makes several referrals to their “approach” which is a bit misleading I think as really the paper demonstrates that FMs can be applied to different datasets.

All references to our "approach" have been reworded

2) The only benchmarks for comparison of performance on the tau tasks are training the model from scratch, this means that a very large data hungry model is often trained on small datasets. It would be informative to know how a smaller model performs on smaller datasets

Added a comparison with the ParT to the text. Using a smaller model (reducing the number of GPT layers) will reduce the performance, as shown on Fig. 7 (scaling with the number of GPT layers).

Addressed the following comments from Reviewer #2:

1- In the introduction in page 3, there is a paragraph about the MPMv1 backbone setup: I think it is quite dense and that it can be omitted as the paper is focusing on OmniJet-α

Removed the detailed description of MPM.

2- Page 5: "from 8192 to 32000 token": This seems a large increase in the token codebook dimension. The authors should motivate this change more it and explain the consequences on the complexity and number of parameters in the backbone model. How does the codebook scale when more features need to be included in the object representation? Does the GPT backbone needs to become much larger to handle such larger codebook?

Added more details regarding the impact of the increased number of tokens.

3- Page 8 "We note that the supervised ParticleTransformer baseline trained specifically for each task outperforms both approaches of using OmniJet-α" how much it outperforms it? The authors should quantify the performance of a baseline model for an easier comparison, e.g. reporting the performance studied in Ref [11]

Added a quantitative comparison between ParT and fine-tuned OmniJet-α for a specific dataset size for all tasks in the paragraph “While using the pre-trained backbone…”.

---

## Editorial Decision

published